

# Implementing deep soil and dynamic root uptake in Noah-MP (v4.5): Impact on Amazon dry-season transpiration

Carolina A. Bieri[1], Francina Dominguez[1], Gonzalo Miguez-Macho[2], and Ying Fan[3]

[1]Department of Climate, Meteorology, and Atmospheric Sciences, University of Illinois Urbana-Champaign, Urbana, Illinois, USA
[2]Nonlinear Physics Group, Faculty of Physics, Universidade de Santiago de Compostela, Santiago de Compostela, Galicia, Spain
[3]Department of Earth and Planetary Sciences, Rutgers University, New Brunswick, New Jersey, USA

**Correspondence:** Francina Dominguez (francina@illinois.edu)

**Abstract.** Plant roots act as critical pathways of moisture from the subsurface to the atmosphere. Deep moisture uptake by plant roots can provide a seasonal buffer mechanism in regions with a well-defined dry season such as the southern Amazon. Most existing state-of-the-art earth system models cannot fully capture the required subsurface-to-atmosphere processes, including groundwater dynamics, a sufficiently deep soil column, dynamic root water uptake, and a fine model spatial resolution (<5

km).

To address this, we present DynaRoot, a dynamic root water uptake (RWU) scheme implemented within the Noah-Multi-Parameterization (Noah-MP) land surface model, a widely used model for studying kilometer-scale regional land surface processes. Our modifications include the implementation of DynaRoot, eight additional resolved soil layers reaching a depth of 20 m, and soil properties that vary with depth. DynaRoot is computationally efficient and ideal for regional- or continental-scale

climate simulations. We perform four 20 year uncoupled Noah-MP experiments for a region in the southern Amazon basin. Each experiment incrementally adds physical processes. The experiments include default Noah-MP with free drainage (FD); addition of a groundwater scheme that resolves water table variations (GW); addition of eight soil layers and soil properties that vary with depth (SOIL), and addition of DynaRoot (ROOT).

Our results show that DynaRoot allows mature forests in upland regions to avoid water stress during dry periods by taking

up moisture from the deep vadose zone (where antecedent precipitation is still draining downward). Conversely, RWU in valleys can take up moisture from groundwater (while remaining constrained by the water table). Temporally, we capture a seasonal shift in RWU from shallower layers in the wet season to deeper soil layers in the dry season, particularly over regions with dominant evergreen broadleaf (forest) vegetation. Compared to the control case, there is a domain-average increase in transpiration of about 29% during dry months in the ROOT experiment. Critically, the ROOT experiment performs best in

simulating the temporal evolution of dry-season transpiration and evapotranspiration (ET) compared with an observational ET product. Future work will explore the effect of the DynaRoot uptake scheme on atmospheric variables in a coupled modeling framework.



## 1 Introduction

The Amazon region of South America is a critical terrestrial source of atmospheric moisture. Evapotranspiration (ET) from
the Amazon basin, defined as the combination of transpiration and evaporation from canopy and ground surfaces, has been
estimated to be about 1100 mm year$^{-1}$ (Baker et al., 2021). Wei et al. (2017) found the ratio of transpiration to ET (excluding
evaporation from canopy interception) to be highest in the Amazon out of any other region in the world, around 85%. ET is
particularly important for Amazonian hydroclimate, as the proportion of precipitation that originates from local ET—known as
the recycling ratio—is estimated to be between 25 and 40% (Dominguez et al., 2022). During the dry season (approximately
June through September), the southern Amazon is a net source of moisture, as evaporation exceeds precipitation (Zemp et al.,
2014). Given the importance of Amazon ET for the water budget of the region, accurate representation of processes which
influence ET in numerical models is critical. More generally, the earth system modeling community has expressed the need for
improved representation of soil hydrology, groundwater variations, and representation of root water uptake (RWU) in global
models (Kleidon and Heimann, 2000; Feddes et al., 2001; Pitman, 2003; Fan et al., 2017; Kendon et al., 2021).

During the dry season, Amazonian canopies maintain greenness (Saleska et al., 2007) and avoid water stress, resulting in
maintenance of dry season ET (Nepstad et al., 1994; Kim et al., 2012; Morton et al., 2014). Previous studies suggest this
is due to several possible mechanisms: nighttime transfer of water from dry to moist areas by roots (known as hydraulic
redistribution; Oliveira et al., 2005), deep RWU (Nepstad et al., 1994), and groundwater capillary rise (Markewitz et al., 2010).
The importance of a particular mechanism could depend on location (Christoffersen et al., 2014) or multiple mechanisms could
operate in synergy (Baker et al., 2009). In this work, we focus on deep RWU given the plentiful evidence for its importance
in areas with seasonal dryness (Nepstad et al., 1994; Jipp et al., 1998; von Randow et al., 2004; Davidson et al., 2011; Ivanov
et al., 2012; Broedel et al., 2017; Fan et al., 2017; Smith and Boers, 2023).

Deep moisture uptake by roots can be thought of as a 'buffer' during seasonally dry periods, allowing vegetation to access
deep vadose zone moisture when surface moisture from precipitation is not sufficient (Nepstad et al., 1994; Jipp et al., 1998;
Baker et al., 2009). Smith and Boers (2023) posited that deep-rooted vegetation is more resilient during dry periods. Miguez-
Macho and Fan (2021) found that 18% of global and annual transpiration originates as deeper soil and rock moisture; in
August, this percentage was estimated to be 60-90% in the southern Amazon. In a modeling study of an artificial throughfall
exclusion experiment at Tapajós National Forest in northern Brazil (Nepstad, 2002; Nepstad et al., 2007; Davidson et al.,
2011), Markewitz et al. (2010) noted that while the percentage of RWU occurring at depths up to 11.5 m was relatively small
(10%), model results suggest it was critical to survival. Thus, while the actual amount of RWU occurring at depth may be small
compared to uptake from shallower depths, it can have an outsized effect on the the vitality of vegetation.

As outlined in Fan et al. (2017) and Miguez-Macho and Fan (2021), topography influences both root access to deep moisture
and the source of this moisture. In very low areas with waterlogging, roots do not grow deep to avoid oxygen stress. A synthesis
of more than 2,000 root observations in Fan et al. (2017) supports this. Moisture flowing downgradient from higher elevations
supports valley vegetation via remote recharge; critically, this source accounts for up to 47% of dry-month RWU (Miguez-
Macho and Fan, 2021). In upland regions where the water table is very deep, roots become decoupled from it, but rely on





moisture storage from previous precipitation (which infiltrates slowly downwards) in the deep vadose zone. It is in midslope and upland areas that we expect roots to tap into deep moisture in the dry season. The influence of drainage gradient on RWU source is summarized in Miguez-Macho and Fan (2021).

Most state-of-the-art earth system models do not currently include an integrated representation of groundwater, deep resolved soil layers and dynamic RWU. Table 1 summarizes existing representation of fine-scale groundwater variations and deep, dynamic RWU in regional land models. Moreover, we consider whether these models can be coupled to a convection-permitting atmospheric model (CPM), since adequate representation of convective precipitation is highly relevant to the hydrological cycle in many parts of the world, including the Amazon (Rehbein et al., 2018). All of the models in Table 1 – with the exception of

Noah-MP and ISAM – were employed in the Coupled Model Intercomparison Project Phase 6 (CMIP6; Eyring et al. 2016). To our knowledge, six of the nine models listed in Table 1 include some representation of deep or dynamic RWU in their official releases, but most do not include both. Only three models (CLM, JSBACH and GFDL LM) include sufficiently deep resolved soil layers to model deep roots in the Amazon. We consider hydrologically active layers extending to 5 m or below to be sufficient based on observations (Restom and Nepstad, 2004; Davidson et al., 2011; Fan et al., 2017). We consider dynamic

RWU to be uptake that varies with time and/or moisture content. Four of the nine models include a groundwater scheme that simulates a lower boundary below the soil column in the form of an aquifer (as opposed to free-draining conditions). One model includes all of the features considered in Table 1, the Community Land Model (CLM) of the Community Earth System Model (CESM; Lawrence et al. 2019). We note that representation of deep, dynamic RWU has been included in model versions that have not been distributed publicly and/or included in an official release. We do not consider such contributions in Table 1,

and instead provide a summary of these efforts in Table 2.

Water table depth variations are best simulated using a fine spatial scale (<5 km) to adequately resolve small-scale topographical features that determine local drainage networks (Fan et al., 2013; Barlage et al., 2021). Advances in groundwater parameterizations – such as the scheme designed and validated in Fan et al. (2007) and Miguez-Macho et al. (2007) – have made it possible to model these fine-scale features. Inclusion of the Miguez-Macho et al. (2007) groundwater scheme (des-

ignated as the Miguez-Macho and Fan, or MMF scheme, for the remainder of this publication) in a model used to simulate regional climate has made it possible to discern links between subsurface moisture variations and land surface fluxes that influence atmospheric heat and moisture budgets, as well as precipitation (Martinez et al., 2016a, b; Barlage et al., 2021).

From Table 1, we see that most existing regional climate models do not offer sufficient functionality to fully resolve the soil-root-atmosphere moisture pathway, neglecting a vital source of moisture for phreatophytic vegetation and potentially

introducing biases in soil moisture, land-atmosphere fluxes and near-surface atmospheric variables. From Table 2, we see that representations of deep, dynamic RWU that do exist are too complex to employ on large spatial scales and/or do not focus directly on deep RWU. To address the need for more representative deep vadose zone hydrology, including RWU, we introduce a modified version of Noah-MP that incorporates three major enhancements: 1) DynaRoot, a RWU scheme described in Fan et al. (2017) that can be seamlessly coupled to deep vadose zone moisture variations and is computationally efficient; 2) an

increase in the number of resolved soil layers from four to twelve with an according increase in cumulative depth from 2 m to



20 m; 3) updated definition of soil properties, which vary with depth based on exponential decay functions without additional input data from the user. Modifications 2 and 3 are necessary for the implementation of DynaRoot.

We implement DynaRoot in the Noah-MultiParameterization model, or Noah-MP (Niu et al., 2011). We select Noah-MP for this work since it is commonly used as a land surface parameterization for the Weather Researching and Forecasting (WRF) model (Skamarock et al., 2021), a widely used numerical weather model ideal for resolving fine spatial scales. Although we focus on uncoupled Noah-MP simulations in this study, in the future we will evaluate the impact of deep RWU on atmospheric variables in a coupled land-atmosphere framework. The WRF model is ideal for simulating the atmosphere on spatial scales fine enough to resolve atmospheric convection. Moreover, we can capture fine-scale variations in water table depth on these smaller scales, allowing us to simulate the important connection between RWU and groundwater variations. The fact that the MMF scheme is already implemented in Noah-MP further supports our decision to implement DynaRoot in this model Barlage et al. (2015); Martinez et al. (2016a, b). Finally, we focus our efforts on Noah-MP because it is one of the three models listed in Table 1 that currently includes no representation of deep or dynamic RWU in its official release. Yet, the Noah-MP/WRF framework is frequently employed in studies of regional climate (Barlage et al., 2015; Martinez et al., 2016a, b; Spera et al., 2018; Fersch et al., 2020; Schwitalla et al., 2020; Barlage et al., 2021; Dominguez et al., 2024).

To focus our work and demonstrate the functionality of our Noah-MP modifications, we test four hypotheses:

– H1: Access to moisture from groundwater is critical for valley vegetation.

– H2: Deep vadose zone moisture is critical for upland vegetation.

– H3: Dynamic root uptake, according to the soil water profile, is most important during the dry season and sustains transpiration.

– H4: Dynamic root uptake is more prevalent and more strongly influences transpiration of mature forests with deeper uptake profiles compared to unforested areas with shallower uptake.

This research is an important step forward towards more physically realistic simulation of the biophysical link between the subsurface and atmospheric branches of the hydrologic cycle in the model. Critically, this framework can be used at spatial scales that are most relevant for land-atmosphere fluxes. These developments can be valuable contributions to the larger Noah-MP and land surface modeling community, and will allow others to more effectively explore science questions regarding the role of vegetation in regional hydroclimate.

## 2 Methods

### 2.1 Description of default Noah-MP

In this study, we use the High-Resolution Land Data Assimilation System (HRLDAS) Noah-MP version 4.5 (https://github.com/NCAR/hrldas/tree/release-v4.5-WRF; Chen et al. 2007), which is consistent with version 4.5 of WRF (https://github.com/wrf-



model/WRF/tree/release-v4.5; Skamarock et al. 2021). We use the model in its default state as the control configuration in the suite of simulations described in section 2.3 of the Methods below.

A common method of Noah-MP initialization is to provide a file generated by the WRF Pre-Processing System (WPS; https://github.com/wrf-model/WPS). Model input files from WRF WPS provide initial values for variables such as soil mois-
ture, soil temperature, and equilibrium water table depth, and also define static variables such as vegetation type and dominant soil texture for the domain. There are several vegetation datasets available in WPS; we use the default 21-class Moderate Resolution Imaging Spectroradiometer (MODIS) land use dataset in all simulations. Outside of the United States, dominant soil texture data in WPS is sourced from the Food and Agriculture Organization (FAO) Soil Map of the World (FAO/UNESCO, 1971).

Default Noah-MP includes four resolved soil layers which extend to 2 m depth. There are several options for determining soil properties in the model. In our control configuration, we employ the default option in which soil properties are determined by the dominant soil texture at a given grid cell and do not vary with depth. Soil moisture at saturation, saturated hydraulic conductivity, saturated hydraulic diffusivity, soil moisture at wilting point, and saturated soil matric potential, among other soil properties, are defined in a lookup table. In default Noah-MP, root depth for a given grid point is also specified via a lookup
table and is based on the dominant vegetation type at that point. The root depth determines the soil layers from which moisture for transpiration is extracted via RWU. It does not change with time or moisture content.

In Noah-MP, RWU for a given soil layer $j$ is calculated as

$$RWU_j = s_j T \tag{1}$$

where $T$ is transpiration at a given grid cell, and

$$
\quad s_j = \beta_j =
\begin{cases}
\frac{\theta_j - \theta_{wilt_j}}{\theta_{ref_j} - \theta_{wilt_j}} & Noah \\
\frac{\psi_{wilt_j} - \psi_j}{\psi_{wilt_j} + \psi_{sat_j}} & CLM \\
1 - e^{-5.8 \ln \frac{\psi_{wilt_j}}{\psi_j}} & SSiB
\end{cases}
\tag{2}
$$

depending on the option for calculation of $\beta_j$ (known as the soil moisture stress factor) set by the user (Niu et al., 2011). Several options are available, including Noah type, CLM type, and SSiB type formulations. $\theta_j$ ($\psi_j$) is soil moisture (soil matric potential) for layer $j$, $\theta_{wilt_j}$ ($\psi_{wilt_j}$) is the wilting point soil moisture (wilting point soil matric potential), $\psi_{sat_j}$ is saturated soil matric potential, and $\theta_{ref_j}$ is the reference soil moisture. See Niu et al. (2011) for further details on the formulation of $\beta_j$
in Noah-MP.

## 2.2 Description of modified Noah-MP

### 2.2.1 Additional soil layers

Soil layer depths as defined in our modified Noah-MP simulations are shown in Table 3. We add eight soil layers, bringing the total number to 12 with a cumulative depth of 20 m. This means hydrologically active soil layers in modified Noah-MP are





deep enough to capture RWU consistent with rooting depths observed in the Amazon (Restom and Nepstad, 2004; Davidson et al., 2011; Fan et al., 2017). This is not the case in default Noah-MP. As WRF WPS only provides initial values for the original four soil layers in default Noah-MP, it is necessary to obtain initial values of soil moisture and temperature for the additional resolved layers in our modified setup. We assigned soil moisture to these layers via a calculated equilibrium profile based on initial water table values from WRF WPS. The details of this process are included in the appendix.

### 2.2.2 Varying soil properties with depth

It is critical to capture observed decreases in porosity and permeability of geologic materials with depth (Fan et al., 2007), particularly in the case of a deeper resolved soil column in our modified Noah-MP setup. Parameterization options that allow varying soil properties with depth are included in Noah-MP, but their use requires additional soil input data for the added layers. These data are not possible to obtain at the depths and spatial resolution we are concerned with. Thus, we implement

exponential decay functions that can describe changes in soil properties with depth on kilometer scales (Miguez-Macho and Fan, 2012):

$$\psi_{sat_j} = \psi_{sat_\circ} \exp \frac{z_j}{f} \qquad (3)$$

$$\theta_{sat_j} = \theta_{sat_\circ} \exp \frac{-z_j}{f} \qquad (4)$$

$$\theta_{wilt_j} = \theta_{wilt_\circ} \exp \frac{-z_j}{f} \qquad (5)$$

$$K_{sat_j} = K_{sat_\circ} \exp \frac{-z_j}{f} \qquad (6)$$

$$D_{sat_j} = \frac{-K_{sat_j}\psi_{sat_j}b}{\theta_{sat_j}} \qquad (7)$$

where $K_{sat_j}$ is saturated soil hydraulic conductivity, $D_{sat_j}$ is saturated soil diffusivity, $f$ is the e-folding depth for permeability, $b$ is the Clapp–Hornberger exponent corresponding to the grid point dominant soil type (Clapp and Hornberger, 1978), and $z_j$ is the depth of soil layer $j$. Variables with a nought subscript indicate the value of that variable in the first soil layer. Details of

the calculation of $f$ are found in Fan et al. (2007) and Miguez-Macho and Fan (2012). $f$ is higher in flat sedimentary basins and lower for steep mountain slopes, which reflects the fact that steep slopes shed sediments while valleys accumulate them. Equation (7) was derived based on Darcy's law describing flow through a porous medium.

As part of the MMF groundwater scheme, a variably thick soil layer is added at grid points where the water table is below the resolved soil layers (Miguez-Macho et al., 2007). This layer extends from the bottom of the lowest resolved layer to the

water table. Soil properties for the variably thick layer are also based on the exponential decay functions shown above, with $z$ set to a constant (22.5 m) for simplicity regardless of the thickness of the layer.

### 2.2.3 The DynaRoot scheme

Following Fan et al. (2017), DynaRoot is analogous to Ohm's law of current flow between two points given a potential difference and conductor resistance. RWU happens preferentially in layers with lower resistance (wetter and shallower layers).





Vegetation relies on RWU from higher-resistance layers (drier and deeper layers) only when the soil water profile "allows" it. In other words, DynaRoot simulates plant behavior of balancing need for water with the effort required for uptake.

The scheme is composed of two main functions. These include an ease function ($e_j$) calculated at each soil layer ($j$) and a fractional contribution to RWU for each soil layer ($r_j$), known as the root activity function:

$$e_j = \frac{\psi_j - \psi_{lmin}}{\frac{2}{3}h_{veg} + z_j} \tag{8}$$

$$r_j = \frac{e_j \Delta z_j}{\sum_{j=1}^{n} e_j \Delta z_j} \tag{9}$$

where $\psi_{lmin}$ is the minimum leaf water potential (set as a constant to -204 m, or -2 MPa), $h_{veg}$ is the vegetation canopy height corresponding to the grid point dominant vegetation type, and n is the number of resolved soil layers. The ease function $e_j$ is based on the concept that RWU is influenced by the local soil water profile, which is determined by both infiltration from above and interactions with groundwater from below. For further information on the conceptual basis of DynaRoot, see the supplementary information provided in Fan et al. (2017). DynaRoot is added as a new module in the energy subroutine in modified Noah-MP. The values of $e_j$ and $r_j$ are calculated in the new module. In the water subroutine of Noah-MP, $r_j$ replaces the beta factor $\beta_j$ for soil moisture stress such that $s_j = r_j$ in Eq. (2).

Moreover, soil layers can be designated as having active or inactive root activity. This is determined by the value of $e_j$. If the value of $e_j$ for layer $j$ is zero for one model year, that layer is flagged as inactive. The layer can only become active again if it is the easiest layer from which to take up moisture; i.e., if the value of $e_j$ exceeds the value of $e_j$ everywhere else in the soil column. Additionally, in modified Noah-MP root depth at a given grid point is allowed to vary in time. Soil layers designated as active comprise the root zone; the root depth is designated as the depth of the deepest layer with active root uptake.

In modified Noah-MP, we calculate the maximum depth of RWU, $D_{RWU}$, as the soil layer $j$ at which $\sum_{j=1}^{n} r_j T >= 0.95T$. If $T$ is zero, $D_{RWU}$ is set to the depth of the first resolved layer (0.1 m). DynaRoot constrains RWU to soil layers above the water table. If the water table is calculated to be above the bottom of the first resolved soil layer, RWU is constrained to that layer.

We do not employ Noah-MP's crop model in any of our simulations to lessen complexity and avoid additional computational cost. Given this, we do not expect our results to be reliable in areas with crop irrigation.

### 2.3 Description of Noah-MP simulations

We ran modified Noah-MP for a domain located in the southern Amazon, within the Brazilian state of Rondônia. Figure 1 depicts the simulation domain within South America (1a), elevation (1b), dominant soil texture (1c), and vegetation cover (1d) as defined in our simulations. We selected a relatively small region that would allow us to carry out a proof of concept for our modified version of Noah-MP. We sought a domain which includes areas designated as needing deep soil water, as has been done in studies such as Nepstad et al. (1994).

Table 4 lists the Noah-MP cases analyzed in this study, known as FD (free drainage), GW (MMF groundwater scheme activated), SOIL (identical to GW but with additional soil layers), and ROOT (identical to SOIL but with DynaRoot activated).



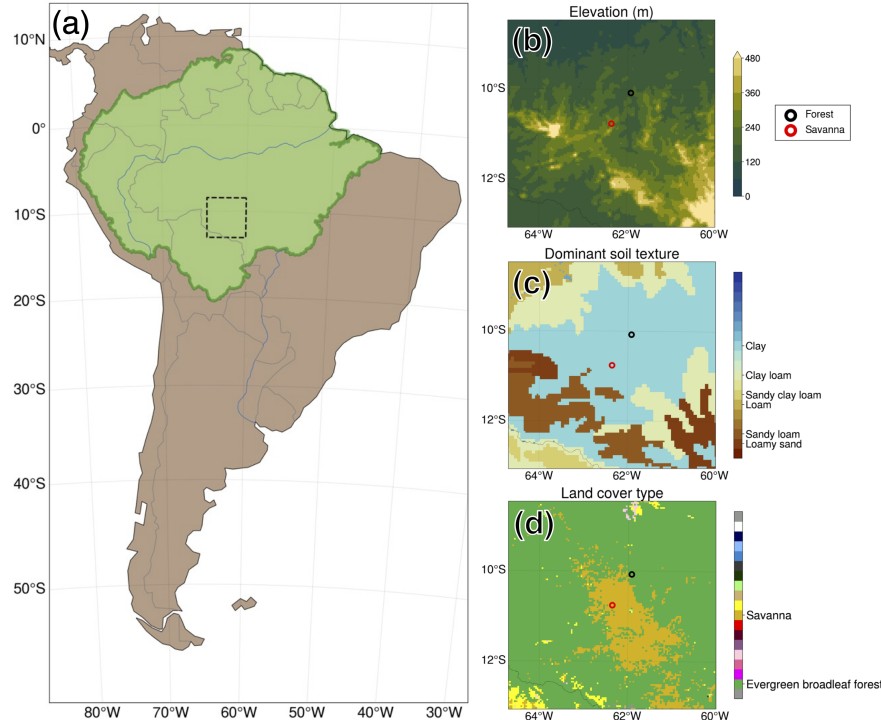

**Figure 1.** Location of Noah-MP simulation domain and static input fields derived from WRF WPS. (a) Location of Noah-MP simulation domain for all cases (black dashed box). The green shaded area denotes the boundaries of the Amazon basin. Made with Natural Earth (public domain). (b) Elevation in the simulation domain. (c) Similar to (b), but for grid point dominant soil texture. (d) Similar to (b) and (c), but for grid point dominant vegetation.

FD acts as the control case in this study. The experimental design allows us to test the hypotheses listed in the Introduction. GW corresponds to H1, enabling us to test the effect of groundwater convergence in valleys on uptake. The SOIL experiment aids in testing H2, with deeper resolved soil layers that can store past precipitation in the deep vadose zone. Finally, the ROOT

experiment simulates vegetation reliance on deep RWU via the DynaRoot uptake scheme, allowing us to test H3 and H4. The incremental nature of the cases (increasing in complexity from FD to ROOT) allows us to isolate the effect of individual physics and modifications.

All simulations were offline, i.e. the model was run in an uncoupled state without active interaction between land and atmospheric components. Rather, 3 hourly atmospheric forcing data with the exception of precipitation were sourced from the

Global Land Data Assimilation System (GLDAS; Rodell et al. 2004; Beaudoing et al. 2020). Precipitation data were sourced from the Integrated Multi-satellitE Retrievals for GPM (IMERG) product (Huffman et al., 2023). Simulations were completed for a 20 year period from 01 July 2000 to 31 December 2019. In all simulations, the model was run at a 4 km resolution and 30 min time step. Model output for the first three years was discarded in our analysis to account for model spin up; about three





years of model integration are required for soil moisture in the deepest layers to stabilize (not shown). Also, the DynaRoot
uptake scheme is not fully active until one year into the model integration. Additional simulation setup details are shown in
Table 5. GLDAS data were processed and converted to the necessary format using scripts provided with the HRLDAS source
code. IMERG precipitation rate data were processed separately; data were averaged into 3 hourly intervals and regridded to
the resolution of the Noah-MP simulations. All simulations were performed on the National Center for Atmospheric Research
(NCAR) Derecho supercomputer (Computational and Information Systems Laboratory, 2024).

## 2.4  Observational analysis

We compared Noah-MP transpiration and ET to gridded data from the Global Land Evaporation Amsterdam Model v3.8a
(GLEAM; Miralles et al., 2011; Martens et al., 2017). GLEAM provides estimates of ET and its components via a set of
algorithms applied to satellite observations (Martens et al., 2017). GLEAM products are available for a continuous 20 year
period that coincides with the analysis period of our Noah-MP runs (2003–2019).

We use the Mann–Kendall test for monotonic trend (Mann, 1945; Kendall, 1948; Gilbert, 1987) to quantify the change in
mean transpiration and ET between days 150 and 250 of the year (the height of the dry season) within our domain of interest.
To quantify the slope of mean change in these variables during this time of year, we use the Theil–Sen slope (Theil, 1950;
Sen, 1968). We compare spatial patterns of the Theil–Sen slope between Noah-MP model runs and the gridded GLEAM
observational product.

# 3  Results

## 3.1  Noah-MP model output

Figure 2 depicts simulation mean water table depth (WTD) and uptake above 1 m in the ROOT experiment for all months, dry
months (Jun-Sep), and wet months (Nov-Feb). Mean WTD is generally deeper in drier months (Fig. 2a), reflecting seasonal
availability of moisture from precipitation. WTD is consistent with simulated values for the same region from other studies
(Martinez et al., 2016a; Fan et al., 2017). Fractional uptake above 1 m (Fig. 2b) varies between dry and wet months, with a
clear shift in uptake to depths below 1 m during dry months. This is consistent with a seasonal shift in RWU from shallower to
deeper areas of the root zone as moisture from precipitation becomes scarce from the wet to dry season.

We examine model output at the point scale to gain an understanding of localized interplay between groundwater, soil
moisture, RWU, and land surface fluxes with and without our Noah-MP modifications. We analyze output for two points in
the domain, a point with dominant evergreen broadleaf forest and a point with dominant savanna vegetation. The locations of
these points are shown in Fig. 1. Given the mean water table locations at these points, we expect uptake to tap into groundwater
capillary rise at the savanna point (in line with H1) and deep vadose zone uptake to occur at the forest point (in line with H2).
We expect dry-season deep RWU to be more relevant at the forest point compared to the savanna point, in accordance with H3
and H4.




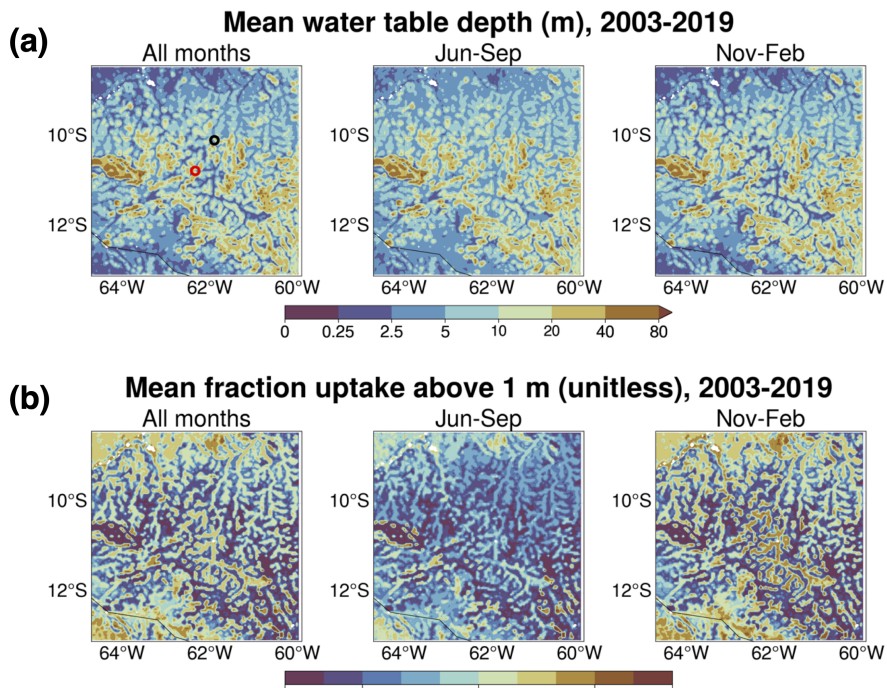

**Figure 2.** Simulation mean results from the ROOT experiment. (a) Simulation mean WTD for all months (left), dry months (Jun-Sep; center), and wet months (Nov-Feb; right). (b) Simulation mean fraction of uptake above 1 m for all months (left), dry months (Jun-Sep; center), and wet months (Nov-Feb; right).

The point-level time series of 1.5 years of model output (Figure 3) support all four hypotheses. In the GW case, locally higher values of soil moisture exist in the vicinity of the water table at the savanna point (Fig 3d, bottom), consistent with capillary rise. The rooting depth at this point (static at 1 m in this experiment) would allow vegetation to tap into this moisture source, consistent with H1. However, there is a disconnect between resolved soil layers (which extend to a cumulative depth of 2 m) and the water table for both forest and savanna points (Figs. 3d, top and 3d, bottom) during all or part of the period. Direct interaction between resolved soil layers and the water table can only occur when the water table is 2 m or shallower. When the water table is deeper than 2 m, indirect interaction occurs via the artificial variably thick layer that extends from the bottom of resolved soil layers to the water table.

In the SOIL experiment, the additional soil layers enable direct interaction between resolved layers and the water table at both points. Periodic decreases in shallow soil moisture occur at the forest point in GW and SOIL (light yellow colors in Figs. 3d and 3e, top and bottom). These periods of decreased soil moisture roughly align with seasonally dry periods (Figs. 3a and 3b, top and bottom). There is a noticeable discontinuity in soil moisture at 2 m depth at the forest point. This is due to the prescribed rooting depth, which is fixed and constrained to 2 m in the SOIL experiment. With the added soil layers in this case, Fig 3e (top) reflects the added soil moisture store in the deep vadose zone, which cannot be fully accessed unless RWU





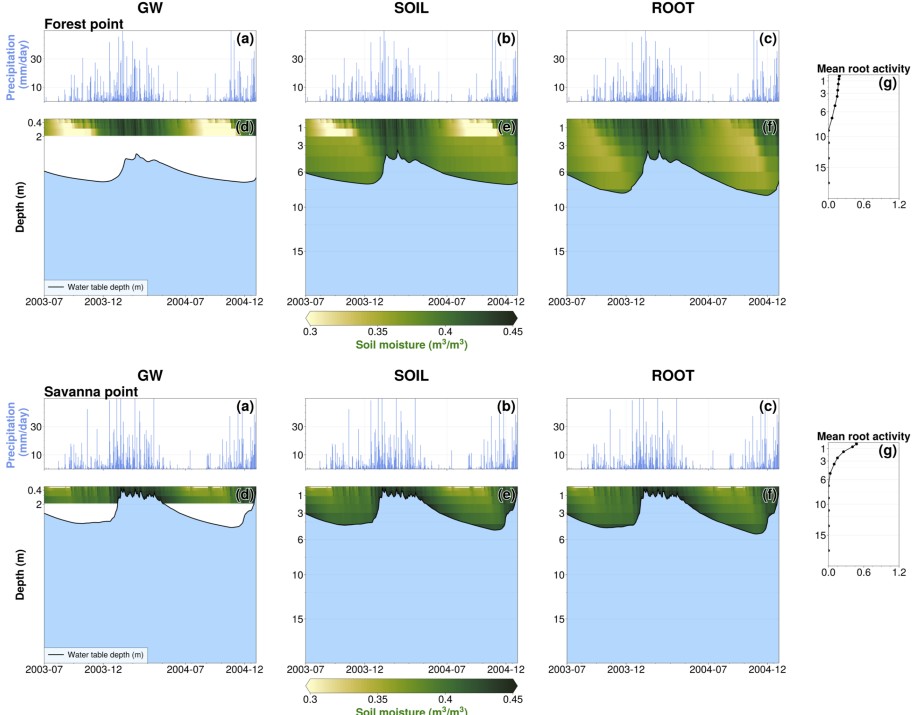

**Figure 3.** Noah-MP model output, 07-2003 to 12-2004. (a)-(c), top row: Mean daily precipitation rate for GW, SOIL, and ROOT experiments at the forest point. (d)-(f), top row: Mean daily soil moisture (shading) and WTD (black line) for GW, SOIL, and ROOT experiments at the forest point. Light blue shading represents regions below the water table. (g), top row: Simulation mean root activity $r_j$ at the forest point. (a)-(g), bottom row: Identical to (a)-(g) in the top row but for the savanna point.

is allowed to extend deeper (H2). At the savanna point, the seasonal depletion of shallow moisture is not as present, signifying
a lack of dependence on root uptake (H4). The vadose zone is not as deep, supporting the accessibility of moisture for uptake
in the capillary fringe (H1).

    In the ROOT experiment – identical to the SOIL setup but with the DynaRoot uptake scheme activated – seasonal soil
moisture changes are more uniformly distributed throughout the soil layers, resulting in less shallow soil drying during the dry
season. This is especially clear at the forest point (Fig. 3f, top). Figure 3g (top) confirms deep and more uniform RWU with
depth at the forest point, as non-zero mean root activity $r_j$ values extend up to the eighth soil layer (which has a layer bottom
that corresponds to 8 m). These results are in support of H2 and H3. When deep moisture is available, vegetation will access
it, particularly in the dry season when surface moisture from precipitation is not sufficient. The source of this moisture is the
deep vadose zone, where moisture from past precipitation is stored. Conversely, Fig. 3g (bottom) confirms less reliance on
deep RWU at the savanna point in the ROOT case, as $r_j$ drops off exponentially from its maximum in the shallowest soil layer.
At the same time, mean root activity at this point indicates that uptake occurs such that vegetation can tap into groundwater




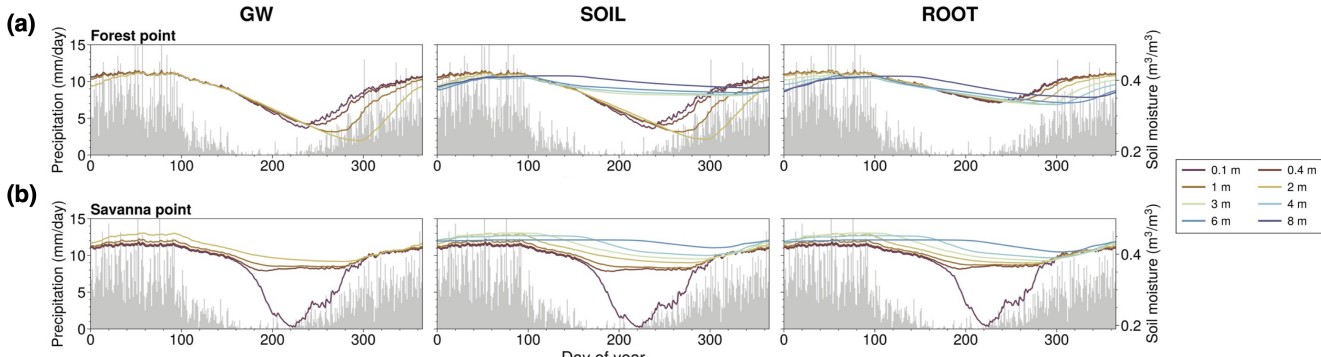

**Figure 4.** (a) Day of year mean soil moisture (multicolored lines) and precipitation (grey bars) for GW, SOIL, and ROOT cases at the forest point. (b) Identical to (a), but for the savanna point. Note that soil moisture is only plotted for layers above the water table.

capillary rise, supporting H1. The mean root activity profiles and soil moisture variations at each point illustrate the dependence of DynaRoot on vegetation type, in support of H4.

Figure 4 shows the mean seasonal cycle of soil moisture for multiple soil layers at the forest point (Fig. 4a) and the savanna point (Fig. 4b) and for GW (left), SOIL (center), and ROOT (right). We also show precipitation for reference. In the GW case

at the forest point (Fig. 4a), soil moisture in all layers is nearly identical until approximately mid-August, when soil moisture in shallower layers begins to rebound from an annual minimum value. Soil moisture in deeper layers continues to decrease before rebounding by late October. At the savanna point (Fig. 4b), soil moisture in deeper layers varies little throughout the year compared to the shallowest layer, which exhibits mostly constant behavior followed by a precipitous decline at the end of the dry season and a sharp rebound. In the SOIL experiment, additional deep soil layers do not affect the behavior of soil

moisture in shallower layers, which exhibit nearly identical behavior to GW at both points. This is due to the fact that rooting depth is consistent with unmodified Noah-MP in this experiment, constrained to a 2 m maximum depth. In the ROOT case, soil moisture in shallow layers shows less of a decline throughout the year at the forest point, while deeper layers show a gradual depletion throughout the dry season compared to SOIL. As such, with the addition of deep soil layers and dynamic RWU, water uptake is more uniformly distributed throughout the soil column during the dry season and transition to the wet season.

At the savanna point, results are relatively unchanged in ROOT, SOIL, and GW. These findings support H3 (dynamic RWU is critical in the dry season) and H4 (dynamic RWU is more important for mature forests compared to non-forested areas).

The behavior suggested by Fig. 4 is further supported by Fig. 5, showing the mean seasonal cycle of RWU in the ROOT case at the forest point (Fig. 5a) and the savanna point (Fig. 5b). For reference, the annual precipitation cycle is depicted in these plots as well. At the forest point, RWU predominantly occurs in deeper layers in the dry season. During other times of

the year, the magnitude of deep RWU is more comparable to RWU in shallower layers. The clear seasonal behavior of RWU exhibited at the forest point is consistent with seasonal dependence on deep RWU in forested areas of the Amazon (Nepstad et al., 1994; Markewitz et al., 2010; Ivanov et al., 2012), consistent with H3 and H4. Notably, RWU in the deepest layer shown





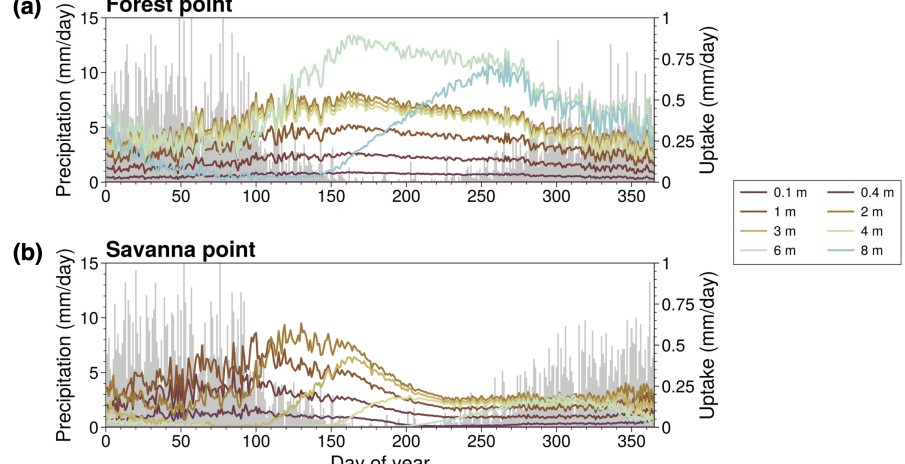

**Figure 5.** (a) Day of year mean RWU (multicolored lines) and precipitation (grey bars) for the ROOT experiment at the forest point. (b) Identical to (a), but for the savanna point.

in Fig. 5a is near zero in the first part of the year and quickly ramps up as the dry season progresses. At the savanna point, RWU predominantly occurs in shallower layers. Seasonal changes in RWU are also evident at the savanna point; RWU peaks

around the end of the wet season and quickly declines through the dry season. This behavior indicates the lack of reliance on RWU at the savanna point compared to forest; RWU is not maintained through the dry season as it is at the forest point. These findings are supportive of H3 and H4.

Annual mean land surface fluxes at the forest and savanna points, including transpiration, latent heat flux, sensible heat flux, ground evaporation, and canopy evaporation, are depicted in Fig. 6. Results for all four cases – FD, GW, SOIL, and ROOT –

are plotted. We show results for the dry season and transition to the wet season (days 130 to 270 of the year) as fluxes do not differ between cases during other parts of the year. In the GW, SOIL, and ROOT experiments at the forest point (Fig. 6, left column), we see an increase (decrease) in transpiration and latent heat flux (sensible heat flux) compared to other simulations from approximately day 160 (early June) through day 270 (mid-October). Importantly, changes in fluxes are only associated with the ROOT experiment which includes DynaRoot. Transpiration increases in accordance with increased availability of

moisture at depth in the ROOT experiment at the forest point (Figs. 3f, top and 4a). This result is in accordance with H3. As there is nearly zero change in canopy or ground evaporation (Figs. 6b and 6c, left), changes in latent heat flux (6d, left) can be attributed to changes in transpiration that result from addition of DynaRoot. This is what we expect given that greater plant access to deep moisture should be reflected in transpiration as opposed to surface or canopy evaporation. Importantly, we note that in the ROOT case, the magnitude of change in surface fluxes increases throughout the dry season and into the transition to

the wet season at the forest point.



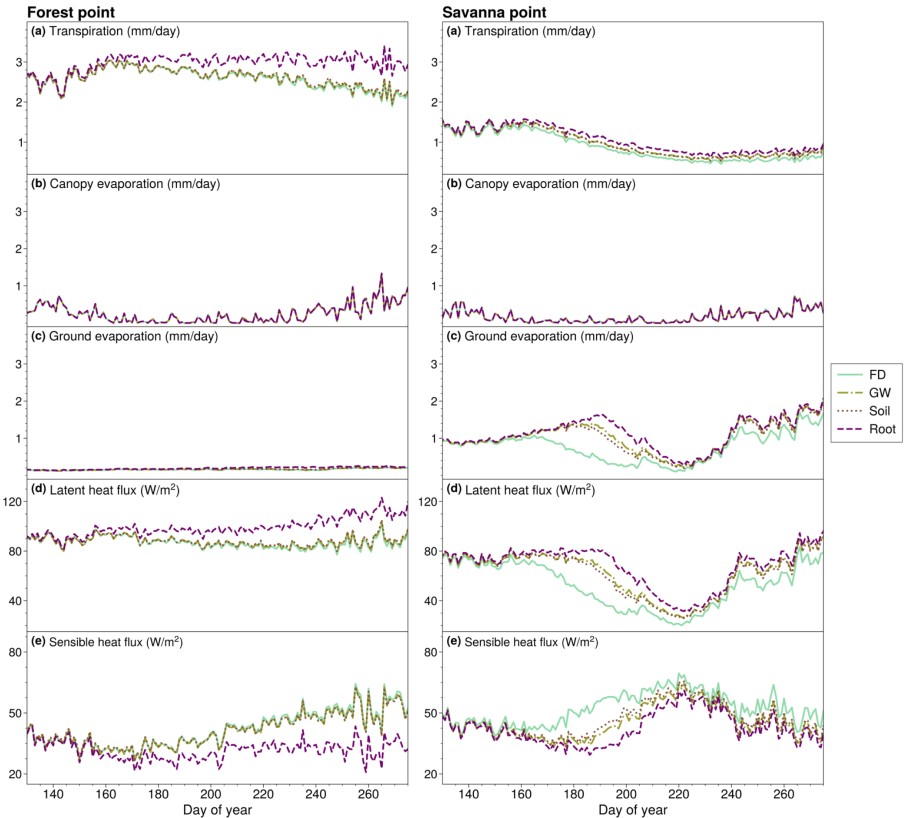

**Figure 6.** (a)-(e), left: Day of year mean (a) transpiration, (b) canopy evaporation, (c) ground evaporation, (d) latent heat flux, and (e) sensible heat flux for FD, GW, SOIL, and ROOT cases at the forest point. (a)-(e), right: Identical to left panel, but for the savanna point.

In contrast to the forest point, the annual mean cycle of dry-season transpiration at the savanna point (Fig. 6a, right column) shows little change throughout the year in all cases compared to FD. Increases in latent heat flux (Fig. 6d) and decreases in sensible heat flux (Fig. 6e) are highest in the early dry season and predominantly occur in the GW and ROOT cases. Given that there is little change in transpiration at the savanna point, changes in latent heat flux must be attributable to other components of surface evaporation. This is shown by the mean annual cycle of dry-season ground evaporation at the savanna point (Fig. 6c, right), which increases in the early dry season and evolves similarly to latent heat flux. We speculate that the increase in early dry-season ground evaporation is tied to increases in soil moisture associated with an active GW scheme in the GW and SOIL cases and more uniform (yet shallow) RWU with depth in the ROOT case. Given the shallow root activity profile at the savanna point (Fig. 3g, bottom) and lack of change in dry-season transpiration (Fig. 6a, right), we deduce that changes in the surface energy balance are not associated with enhanced uptake as a result of adding DynaRoot. These results support H4.

We present a regional analysis to highlight geographical variations within the domain. Figure 7 depicts the seasonal mean difference in transpiration between FD and all other simulations averaged over all years of model output. As expected from the



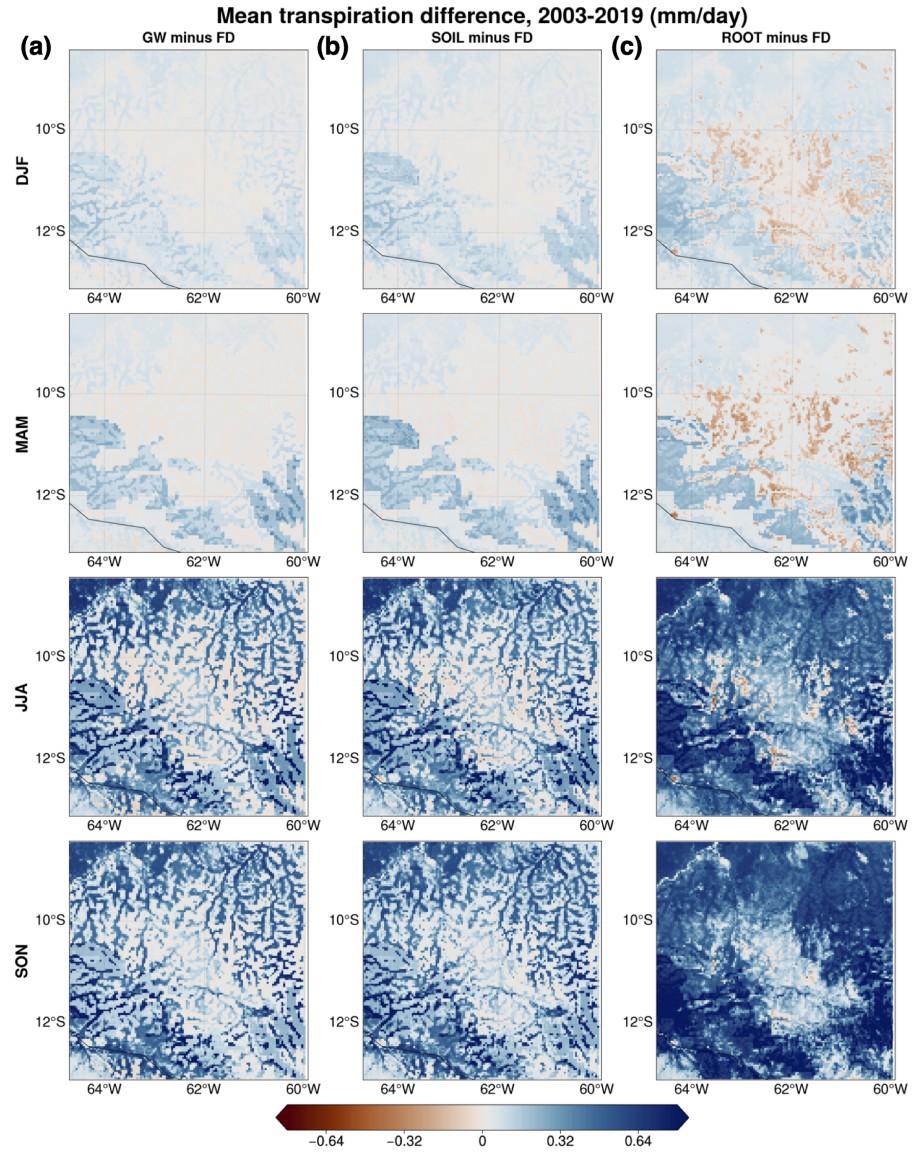

**Figure 7.** Seasonal mean transpiration differences between (a) GW and FD, (b) SOIL and FD (center), and (c) ROOT and FD.

point analysis, differences are minimal during wetter seasons (DJF and MAM) and largely positive during drier seasons (JJA and SON). In SON, when the greatest difference between runs is calculated, the domain average percent difference between the ROOT and FD experiments is about 29%, corresponding to an increase in domain mean transpiration of 0.46 mm day$^{-1}$.

Regarding the spatial distribution of transpiration differences in Fig. 7, there is a visible pattern consistent with mean WTD simulated by the model (Fig. 2). Critically, transpiration differences between GW and FD (Fig. 7a) in JJA and SON are mostly in lower-lying areas, where we expect moisture from groundwater to influence soil moisture and in turn, transpiration Miguez-





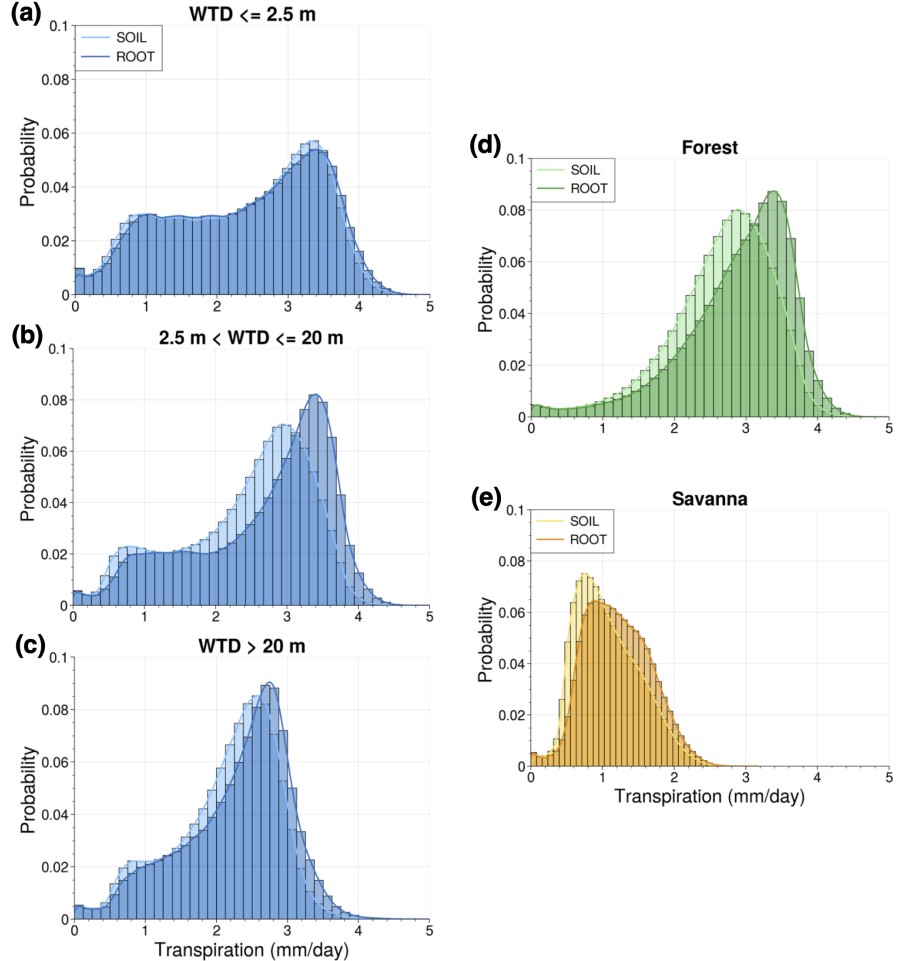

**Figure 8.** Probability density functions depicting distribution of SOIL and ROOT mean dry-season transpiration for different grid points: (a) Points with mean shallow WTD ($<=$2.5 m) in the dry season, (b) points with mean dry-season WTD between 2.5 m and 20 m, (c) points with mean dry-season WTD below 20 m, (d) points with evergreen broadleaf forest dominant vegetation, and (e) points with savanna dominant vegetation.

Macho and Fan (2021). This result supports H1. Transpiration differences between ROOT and FD (Fig. 7c) in JJA and SON are
non-negligible in areas with a higher drainage position, suggesting that upland vegetation tap into moisture in the deep vadose zone when DynaRoot is activated. This finding is consistent with H2 and H3. Moreover, differences in transpiration reflect the dominant vegetation cover at each grid point (Fig. 1); differences are generally larger for forested grid points in agreement with the point analysis and H4.

We calculated probability density functions (PDFs) of mean dry-season transpiration for the simulation with only deep soil
(SOIL) and the simulation with deep soil and DynaRoot (ROOT). We analyze points with a mean water table that is 2.5 m or





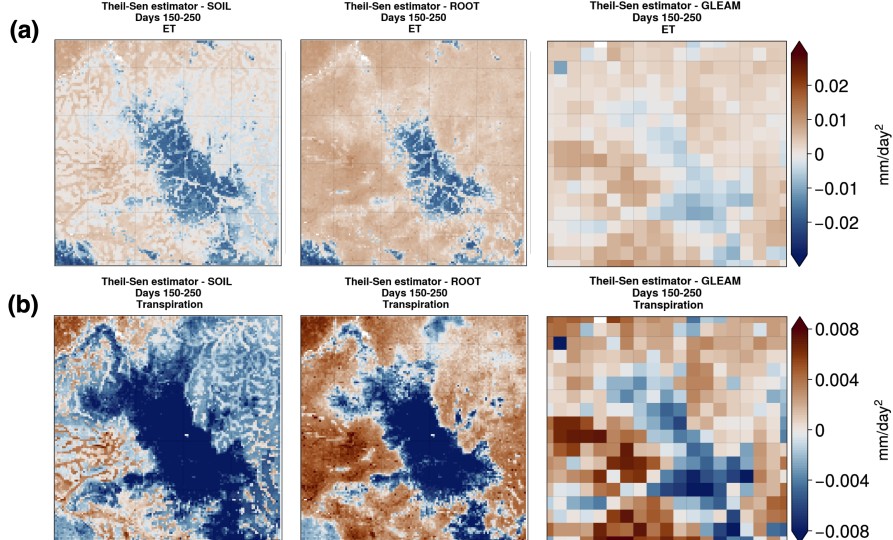

**Figure 9.** (a) Theil-Sen slope for mean ET and (b) transpiration between days 150 and 250 of the year for the SOIL (left column) and ROOT (middle column) Noah-MP experiments and GLEAM (right column) transpiration and ET.

less in the dry season, points with a dry-season mean water table between 2.5 m and 20 m, and points with a dry-season mean water table below 20 m. These PDFs are shown in Fig. 8. We expect to see the DynaRoot uptake scheme mainly impacting midslope (Fig. 8b) and upland points (Fig. 8c) where uptake from the deep vadose zone is important (in accordance with H2). This is indeed what we see in Figs. 8a-8c, with the activation of DynaRoot resulting in a more noticeable shift towards
higher values of dry-season transpiration in Figs. 8b and 8c compared to Fig. 8a. This is particularly true for Fig. 8b, which corresponds to points with mean dry-season WTD between 2.5 m and 20 m.

Additionally, we calculated PDFs of mean dry-season transpiration for points with evergreen broadleaf dominant vegetation (forested) and savanna dominant vegetation (non-forested; Figs. 8d and 8e). We see a greater impact at forested points where LAI is higher and we expect vegetation to be more dependent on deep RWU during the dry season. While there seems to be
somewhat of a shift towards higher values of dry-season transpiration for savanna points (Fig. 8e), there is a much clearer shift for forested points (Fig. 8d). This result is supportive of H4.

### 3.2   Comparison to observations

One of the difficulties in evaluating our model results via comparison to observations is the very limited number of eddy covariance observations in the region, and large uncertainties in ET estimates using remote sensing (Baker et al., 2021). As
an alternative to direct model comparison, we can evaluate how the addition of DynaRoot affects dry-season dynamics when compared to observations. Dry season behavior reflects the resilience of forest vs. savanna vegetation during periods of water stress, and will have an important effect on sub-seasonal surface flux interactions with the overlying atmosphere.





To assess the ability of modified Noah-MP to represent dry-season transpiration and ET dynamics, we perform a comparison with GLEAM data. In Fig. 9a, we see that the Theil–Sen slope (Theil, 1950; Sen, 1968) for ET is generally negative for non-forested areas with savanna and positive for forested areas with evergreen broadleaf vegetation in Noah-MP. This is especially true in the ROOT experiment. We see that with the addition of DynaRoot the sign of the Theil–Sen slope for ET exhibits more agreement with GLEAM than results from the SOIL experiment.

Focusing on transpiration in Fig. 9b, we see that, generally speaking, negative Theil–Sen slopes are calculated for model grid points with savanna and positive slopes are calculated for forested points. Exceptions to this are present in the eastern part of the domain in the SOIL experiment. This is where vegetation is largely classified as evergreen broadleaf forest, but the Theil–Sen slope is generally negative. Comparing with slopes calculated from GLEAM transpiration data, we see that results from the ROOT experiment are more in line with observational estimates. Overall, we can glean something critical from Fig. 9: Addition of the DynaRoot uptake scheme results in dry-season evolution of transpiration and ET that is more in line with observational estimates than a Noah-MP configuration without it. This highlights the role of DynaRoot in realistically simulating dry-season surface flux dynamics at the land-atmosphere interface. Moreover, this result is in support of H3, as we show that inclusion of deep, dynamic RWU is necessary to realistically model the mean evolution of dry-season transpiration for this domain.

## 4 Discussion

The results of this work reflect a major advancement in the representation of the link between subsurface and atmospheric fluxes of moisture via RWU in Noah-MP. The structure of the model in its default state made it impossible to represent time-varying moisture uptake from subsurface sources below 2 m depth, a critical source of dry-season moisture in Amazon forests. Now, we have a way of representing seasonal reliance of vegetation on deep RWU. Moreover, with the availability of an active groundwater scheme in Noah-MP, we can study the interplay between groundwater, soil moisture, roots, and land surface fluxes. This allows us to more fully resolve moisture pathways from the subsurface to the atmosphere, enabling future studies that continue to investigate the role of deep RWU in modulating vegetation water stress and determining atmospheric moisture availability. This is especially important as the Amazon is projected to experience drier conditions in the future as a result of climate change (Joetzjer et al., 2013) and the ecosystem's drought resilience may be at risk (Chen et al., 2024).

Additionally, in this work we have described an approach to account for variations in soil properties with depth. Such properties are constant through the soil profile in unmodified Noah-MP unless the user provides input soil texture data for each layer. We incorporate exponential decay functions that describe changes in soil properties with depth without the need for additional user-provided data. This enhances the realism of the model with minimal overhead and facilitates simulations with additional deep soil layers.

We focus on the Amazon region in this work because of its critical importance for global climate (Werth and Avissar, 2002; Liu et al., 2023). However, our modified model would be useful in simulating any water-limited ecosystem where vegetation has adapted to periodic water stress. The simplicity of DynaRoot is ideal for coupling with atmospheric models with high



computational overhead such as WRF, providing a realistic approach to accounting for critical land surface and subsurface processes in a coupled land–atmosphere modeling framework without added parameters or calibration.

We note that the DynaRoot uptake scheme does not consider the role of carbon availability in determining RWU depth and the presence of roots; rather, it depends solely on the soil water profile. While the role of carbon is certainly relevant to studies focused on directly simulating root growth based on principles of plant physiology and resource acquisition, here we focus on

hydrologically based rooting depth, allowing the soil water profile to drive root growth. This is sufficient for the scope of our work, without the use of an active dynamic vegetation module that would predict resource-based rooting growth. We elect to run simulations without such a module in the interest of minimizing complexity and computational cost.

## 5 Conclusions

This work details implementation of a deep, dynamic RWU scheme known as DynaRoot in the Noah-MP land surface model.

The conceptual basis of this scheme is outlined in Fan et al. (2017). The scheme determines soil layers with active RWU based on the local soil water profile. It is designed to be conceptually and computationally simple so it can be employed on large spatial scales and/or in high-resolution land–atmosphere modeling frameworks. Despite its simplicity, the scheme provides a way to capture seasonal dependence of mature forests on deep RWU in the Amazon and other seasonally dry ecosystems. This is critical behavior that prevents water stress during dry periods. Previously, it was not possible to capture deep, dynamic RWU

in Noah-MP. Importantly, we are now able to obtain a measure of RWU depth based on the local soil water profile. Previous studies have illustrated the crucial reliance of Amazonian forests on deep RWU via observational (Nepstad et al., 1994; Broedel et al., 2017) and modeling-based (Kleidon and Heimann, 2000; Markewitz et al., 2010) methods.

In addition to implementing DynaRoot, we modified other parts of the Noah-MP code to allow for simulation of deep RWU. First, we included eight additional resolved soil layers that extend to a depth of 20 m in place of the traditional Noah-MP

setup, with four layers that extend to 2 m depth. To initialize values of soil moisture and temperature, we added methods that estimate values of these variables based on water table depth and deep soil temperature from model input data generated with WRF WPS. Moreover, we included modifications to soil properties to capture changes in porosity and permeability of the soil column with depth; we use exponential functions that were utilized in Fan et al. (2007) to do this.

We completed several offline Noah-MP simulations for a test domain in the southern Amazon. These cases include an

out-of-the-box unmodified Noah-MP case with free drainage (FD), which acts as a control configuration, an unmodified Noah-MP case with a groundwater scheme activated (GW), a modified Noah-MP case with additional resolved soil layers and soil properties that vary with depth (SOIL), and a modified Noah-MP case identical to SOIL but with the DynaRoot uptake scheme activated (ROOT). Each case facilitates testing of a different hypothesis. Addition of an active groundwater scheme in GW (and comparison with FD) allows us to test H1: moisture from groundwater is critical for valley vegetation. Addition of deep

resolved soil layers in the SOIL case facilitates testing of H2: deep vadose zone moisture is critical for upland vegetation. Activation of DynaRoot in the ROOT case allows us to test H3: dynamic RWU is important in the dry season and sustains transpiration, and H4: dynamic RWU is strongly influences transpiration of mature forests compared to non-forested areas.





We analyzed output from these simulations at the domain scale and at the point scale. At the point scale, we examined output for a forested grid point with a deeper mean WTD and a savanna (non-forested) grid point with a shallower mean WTD.

Selection of these points facilitates testing of all four hypotheses. Even with explicit representation of groundwater in the GW case, it is not possible to capture interactions between a water table below 2 m and resolved soil layers because the resolved layers are too shallow. The SOIL case remedies this by including the eight additional soil layers, representing deep vadose zone storage at the forest point. In the ROOT case, addition of DynaRoot allows model rooting depth to vary in time with local soil water availability; as a result, soil moisture is depleted more uniformly as RWU shifts to the deep vadose zone during the

dry season. This behavior is most apparent at the forest point, supporting H2, H3, and H4. Capillary rise in the vicinity of the water table is modeled at the savanna point, and modeled mean root activity indicates uptake sourced from the capillary fringe, consistent with H1. The seasonal cycle of RWU indicates deeper RWU at the forest point during the dry season compared to the savanna point (this further supports H4). At the savanna point, RWU predominantly occurs in shallower layers and maximizes early in the dry season, then quickly declines. Forest dry-season transpiration and latent heat flux (sensible heat flux) increase

(decreases) considerably as a result of adding DynaRoot, signifying a change in surface energy flux partitioning. Changes in transpiration at the savanna point with addition of DynaRoot are minimal. Changes in latent heat flux at the forest point are mostly accounted for by changes in transpiration, while changes in latent heat flux at the savanna point are accounted for by changes in ground evaporation likely associated with increased near-surface soil moisture (and not deep RWU). Changes in surface fluxes are in accordance with H3 and H4. In this point-level analysis, we capture differing behavior between forested

and non-forested vegetation in the dry season, and obtain a mechanistic view of our Noah-MP modifications.

The domain scale analysis suggests our point scale results are valid for larger areas. The domain scale analysis shows that changes in seasonal mean transpiration for the entire simulation period are maximized during the dry months (JJA and SON), for forested areas, and in the case with an active RWU scheme (ROOT), in support of H3 and H4. Moreover, our comparison of dry-season transpiration in the GW case to FD supports H1; increases in transpiration predominantly occur in valleys when a

groundwater scheme is activated. Domain-averaged increase in transpiration between the ROOT and FD experiments is about 29%, corresponding to an increase in transpiration of 0.46 mm day$^{-1}$. When adding deep, dynamic RWU, the distribution of mean dry-season transpiration shifts towards higher values in areas where the mean dry-season water table is between 2.5 m and 20 m or deeper than 20 m (as opposed to areas with a shallower mean WTD), and for forested grid points (as opposed to non-forested grid points). These findings are consistent with H2, H3, and H4. In the ROOT experiment, addition of DynaRoot

facilitates greater deep vadose zone moisture uptake. Enhanced deep uptake enables increased transpiration in the dry season in forested regions.

Comparison of our simulation results with GLEAM remote sensing estimates of dry-season moisture fluxes for the domain reveal that dry-season temporal evolution of ET and transpiration in forested areas better agrees with GLEAM in the ROOT case (DynaRoot activated) compared to the SOIL case (DynaRoot not activated). These results confirm the value of our Noah-

MP modifications in realistically capturing seasonal moisture flux dynamics at the land-atmosphere interface in an ecosystem that is dependent on deep RWU to buffer rainfall deficits. Moreover, the results of this comparison provide further support for H3.





Overall, we find that the results of this work support the hypotheses detailed in the Introduction and are in accordance with previous studies that motivated these hypotheses. These include Fan et al. (2017), which highlighted the importance of
groundwater as a moisture source for vegetation during dry periods, and Miguez-Macho and Fan (2021), which clarified that while moisture from groundwater is important in valleys, deep vadose zone storage of past precipitation is critically important in uplands during dry months.

Critically, the changes to Noah-MP outlined in this work mean we now have a numerical model that can capture the spectrum of interactions from groundwater through deep soil and plant roots to the atmosphere via transpiration. To our knowledge,
no existing land-atmosphere modeling framework includes sufficient representation of all these components to capture the influence of groundwater and the local soil moisture profile on rooting depth for convection-permitting scale modeling.

Future work will focus on further validating these efforts by comparing model results with remote sensing observations which can provide more coverage in space and time than existing transpiration observations. An additional goal in the future is to complete coupled Noah-MP and WRF simulations that will allow us to understand potential effects of the DynaRoot
uptake scheme on simulation of atmospheric variables. As we strive to understand the state of our changing world, it is more imperative than ever to adequately characterize vegetation and its influence on hydroclimate in critical ecosystems such as the Amazon.

*Code and data availability.* The version of HRLDAS Noah-MP used in this study is available at https://doi.org/10.5281/zenodo.13137185. Model configuration, input, and forcing files are available at https://doi.org/10.5281/zenodo.13061970. Scripts used to process, analyze, and
plot model and observational data are available at https://doi.org/10.5281/zenodo.13137808.

Atmospheric forcing data used for the model simulations in this study are publicly available online. GLDAS data are available from https://ldas.gsfc.nasa.gov/gldas. IMERG data are available from https://gpm.nasa.gov/data/imerg. Static input data used for the model simulations are available at https://www2.mmm.ucar.edu/wrf/users/download/get_sources_wps_geog.html. The base HRLDAS Noah-MP source code and documentation are available at https://github.com/NCAR/hrldas. WRF WPS source code and documentation are available at
https://github.com/wrf-model/WPS. GLEAM ET and transpiration data used in this study are available at https://www.gleam.eu. The Amazon basin shapefile used to generate Fig. 1 is available at https://github.com/gamamo/AmazonBasinLimits.

## Appendix A: Initialization procedure for additional soil layers

Initial soil moisture values for layers 5 through twelve were derived using a formulation of the Richards equation describing water flow in unsaturated soils:

$$q_j = \frac{1}{2} D_{sat_j} \frac{(\theta_j - \theta_{j+1})}{\Delta z_j} + K_{sat_j} \left(\frac{\theta_j}{\theta_{sat_j}}\right)^{b+1} \tag{A1}$$

where $q_j$ is the flux of water between a soil layer $j$ and the layer below it $j+1$, $D_{sat_j}$ is saturated soil hydraulic diffusivity, $\theta_j$ is soil moisture in soil layer $j$, $\Delta z_j$ is the thickness of soil layer $j$, $K_{sat_j}$ is saturated hydraulic conductivity, $\theta_{sat_j}$ is soil



moisture at saturation, and $b$ is the Clapp–Hornberger exponent corresponding to the grid point dominant soil type (Clapp and Hornberger, 1978). We assume $q_j$ to be zero at the initial time step for simplicity.

Equilibrium WTD, which is sourced from the input file generated by WRF WPS, is used for the initial WTD values. The details of how the equilibrium WTD data were created can be found in Fan et al. (2007). Given the equilibrium WTD at a given grid point, we can infer $\theta_j$ at each layer by starting at the layer $j+1$ with the water table and iterating upwards. $\theta_{j+1}$ can be determined by considering equilibrium soil moisture $\theta_{eq_{j+1}}$ (soil moisture corresponding to the scenario in which the water table is located exactly at the bottom of the soil layer, previously calculated in the model) and $\theta_{sat_{j+1}}$:

$$\theta_{j+1} = \theta_{sat_{j+1}} \frac{WTD - z_{j+1}}{\Delta z_{j+1}} + \theta_{eq_{j+1}} \frac{z_j - WTD}{\Delta z_{j+1}}. \tag{A2}$$

where $z_j$ and $z_{j+1}$ are the depths of soil layers $j$ and $j+1$. From there, $\theta_j$ can be estimated by solving Eq. (A1) iteratively using the Newton–Raphson method, a numerical method that solves for the root of a well-behaved function (Press et al., 1992). In the next iteration, $\theta_j$ replaces $\theta_{j+1}$ and the new $\theta_j$ corresponds to the next layer (moving upwards). This continues until initial soil moisture values for remaining layers (until layer 4) are calculated.

Initial soil temperature values were obtained by linear interpolation. Initial values for the first 4 layers were provided by the WRF WPS input file, as well as a deep soil temperature initialization value corresponding to 8 m depth. We took this value as representative of 20 m depth (the bottom of the soil column in our modified Noah-MP simulations; F. Chen, personal communication, August 16, 2021), and estimated initial values for soil temperature in the additional layers based on this value and the initial value at the fourth layer (2 m depth).

*Author contributions.* C.A.B. prepared the manuscript with contributions from all co-authors. C.A.B. carried out the Noah-MP code modifications with contributions from G.M.M. The DynaRoot scheme was developed by Y.F. and G.M.M. C.A.B. completed simulations and analyses with contributions from F.D. F.D., Y.F. and G.M.M. were responsible for conceptualization of the study. F.D. supervised execution of the study and preparation of the manuscript.

*Competing interests.* The authors declare that they have no conflict of interest.

*Acknowledgements.* This material is based upon work supported by the National Science Foundation under Grant No. 1852707 and Grant No. 1852709. Any opinions, findings, and conclusions or recommendations expressed in this material are those of the author(s) and do not necessarily reflect the views of the National Science Foundation. C.A.B. gratefully acknowledges financial support for this research by the Fulbright U.S. Student Program, which is sponsored by the U.S. Department of State and the U.S.-Spain Fulbright Commission. The contents of this publication are solely the responsibility of the author and do not necessarily represent the official views of the Fulbright Program, the Government of the United States, or the U.S.-Spain Fulbright Commission. C.A.B. gratefully acknowledges financial support from the





Alfred P. Sloan Foundation's Minority PhD Program. The HRLDAS modeling system was developed at the National Center for Atmospheric Research (NCAR). NCAR is sponsored by the United States National Science Foundation.



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

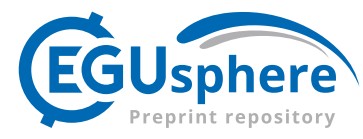

**Table 1.** Summary of representation of deep, dynamic RWU, groundwater (GW) dynamics, and capability of CPM coupling in existing land surface models. We consider deep RWU to be RWU that occurs at 5 m or below. We consider dynamic RWU to be RWU that changes with time and/or moisture content.

| Model | Number of soil layers | Cumulative depth of soil layers | Deep RWU? | Dynamic RWU? | GW scheme? | Couple with CPM? | Relevant publications |
|---|---|---|---|---|---|---|---|
| Noah-Multiparameterization Model (Noah-MP) | 4 | 2 m | No | No | Yes | Yes | Niu et al. (2011); He et al. (2023) |
| Integrated Science Assessment Model (ISAM) | 15 | 50 m (total); 3.5 m hydrologically active | No | Yes | No | Yes | Jain et al. (2009); Song et al. (2013); Song et al. (2016); Lin (2022); Lin et al. (2023) |
| Community Land Model (CLM) | 25 | 5̃0 m (total); 8.5 m hydrologically active | Yes | Yes | Yes | Yes | Lawrence et al. (2019); Kennedy et al. (2019) |
| Energy Exascale Earth System Model (E3SM) Land Model (ELM) | 15 | 42.1 m (total); 3.8 m hydrologically active | No | Yes | Yes | Yes | Oleson et al. (2013); Bisht et al. (2018); Drewniak (2019); Caldwell et al. (2021); Golaz et al. (2022); Qiu et al. (2023) |
| Jena Scheme for Biosphere-Atmosphere Coupling in Hamburg (JSBACH) | 5 | 9.8 m | Yes | No | No | Yes | Reick et al. (2021); Schneck et al. (2022); Bao et al. (2024) |
| Revised Hydrology for the Tiled ECMWF Scheme for Surface Exchanges over Land (HTESSEL) | 4 | 2.89 m | No | No | No | No | Balsamo et al. (2011); ECMWF (2023) |
| Organizing Carbon and Hydrology in Dynamic Ecosystems (ORCHIDEE) | 12 | 4 m | Yes | Yes | No | Yes | Krinner et al. (2005); Verbeeck et al. (2011); Peng et al. (2014); Naudts et al. (2015); Yao et al. (2022); Shahi et al. (2022); Joetzjer et al. (2022) |
| Joint UK Land Environment Simulator (JULES) | 4 | 3 m | No | No | No | Yes | Best et al. (2011); Batelis et al. (2020); Jucker et al. (2020); Wiltshire et al. (2020) |
| Geophysical Fluid Dynamics Laboratory (GFDL) Land Model (LM) | Variable; 20 in Milly et al. (2014) | Variable; 10 m in Shevliakova et al. (2024) | Yes | Yes | Yes | No | Milly et al. (2014); Shevliakova et al. (2024) Cusack et al. (2024) |





**Table 2.** Summary of existing representations of deep, dynamic RWU in land surface models and how they compare with our approach.

| Reference | Model used | Deep RWU? | Dynamic RWU? | Difference from our study |
|---|---|---|---|---|
| Gayler et al. (2014) | Noah-MP | No | Yes | We seek to implement a lower-complexity scheme that can be easily scaled up to a continental domain |
| Wang et al. (2018) | Noah-MP | Yes | Yes | |
| Niu et al. (2020) | Noah-MP | No | Yes | |
| Li et al. (2021) | Noah-MP | Yes | Yes | |
| Zanin (2021) | Noah (coupled with Eta/CPTEC regional climate model) | Yes | No | We place more emphasis on the role of drainage gradient |
| van Oorschot et al. (2021) | HTESSEL | No | Yes | We focus on enhancing RWU directly |

**Table 3.** Soil layer depths as defined in modified Noah-MP simulations (our SOIL and ROOT experiments). Layers in bold are identical to the layers in unmodified (default) Noah-MP (our FD and GW experiments).

| Layer number | Depth of layer bottom (m) |
|---|---|
| **1** | **0.1** |
| **2** | **0.4** |
| **3** | **1** |
| **4** | **2** |
| 5 | 3 |
| 6 | 4 |
| 7 | 6 |
| 8 | 8 |
| 9 | 10 |
| 10 | 12 |
| 11 | 15 |
| 12 | 20 |



**Table 4.** Noah-MP cases analyzed in this study.

| Simulation name | Description |
| --- | --- |
| FD (Control) | Unmodified Noah-MP run with free drainage groundwater option (RUNOFF_OPTION=3) |
| GW | Unmodified Noah-MP run with the Miguez-Macho and Fan (MMF) groundwater scheme (RUNOFF_OPTION=5; Miguez-Macho et al. 2007) |
| SOIL | Modified Noah-MP with additional soil layers and soil properties varying with depth |
| ROOT | Identical to SOIL but with DynaRoot activated |





**Table 5.** Technical specifications and selected Noah-MP settings for all simulations. A copy of the namelist file used in our simulations is provided with our modified Noah-MP code (see code availability section below).

| | |
|---|---|
| Simulation length | Jul 2000 to Dec 2019 (about 20 years) |
| Horizontal resolution | 4 km |
| Forcing time step | 3 h |
| Model time step | 30 min |
| Atmospheric forcing data (Temperature, wind speed, incoming shortwave radiation, incoming longwave radiation, surface pressure, and specific humidity) | GLDAS V2.1 Level 4 Noah Land Surface Model 3 hourly $0.25° \times 0.25°$ product (Beaudoing et al., 2020) |
| Atmospheric forcing data (Precipitation) | GPM IMERG Final Precipitation L3 Half Hourly $0.1° \times 0.1°$ V07 product (Huffman et al., 2023) |
| Initialization file | WRF WPS input file |
| DYNAMIC_VEG_OPTION | Dynamic vegetation model inactive; use vegetation fraction from input data and monthly-mean LAI (Option 7) |
| CANOPY_STOMATAL_RESISTANCE_OPTION | Ball-Berry (Ball et al., 1987) (Option 1) |
| BTR_OPTION | Noah-type (Niu et al., 2011) (Option 1) |
| SURFACE_DRAG_OPTION | Monin-Obukhov (Option 1) |
| TBOT_OPTION | Temperature at soil bottom (TBOT) read from file (Option 2) |
| SURFACE_RESISTANCE_OPTION | Sakaguchi and Zeng (2009) (Option 1) |