# Peer review of "Implementing deep soil and dynamic root uptake in Noah-MP (v4.5): Impact on Amazon dry-season transpiration"

_EGUsphere, 2024_

## Author Comment (AC1)

We thank Anonymous Referee #1 (AR1) for their helpful feedback on our manuscript. AR1's comments are copied below in black with our responses to each in red.

Deep savanna roots:

The authors focus on deep roots for evergreen broadleaf forests, while the savanna roots are assumed to be shallow. In Singh et al. (2020) the presence of deep roots in savanna regions is highlighted. It is recommended to discuss how your analyses relate to this opposing result.

Singh, Chandrakant, et al. "Rootzone storage capacity reveals drought coping strategies along rainforest-savanna transitions." Environmental Research Letters 15.12 (2020): 124021.

Thank you for bringing up this important point. The findings of Singh et al. (2020) and similar works should be considered. The following has been added to the manuscript (L264-270) , Results section):

*Note that deep-rooted woody vegetation has been shown to exist in savanna ecosystems (Canadell et al., 1996; Oliveira et al., 2005a; Singh et al., 2020). Most of the areas classified as savanna in our domain are a result of deforestation, with forest giving way to grass-dominant pastures with a shallower rooting depth (Gash and Nobre, 1997; Roberts et al., 2002; Von Randow et al., 2004; Piontekowski et al., 2019; Honey, 2023). Additionally, Noah-MP does not capture heterogeneity in growth form within a given vegetation class, making it impossible for us to account for the proportion of potentially deep-rooted woody vegetation in savanna. A single canopy height range is assumed for every grid point classified as savanna (minimum 0.1 m, maximum 10 m) and forest (minimum 8 m, maximum 20 m). This canopy height is considered in calculation of $e_j$.*

Please let us know if you have any follow-up comments on this issue.

Comparison to GLEAM:

The authors compare their model results with the GLEAM evaporation product. It should be noted that the 'algorithms applied to satellite observations' (L233) include assumptions on roots and root water uptake. Given this, referring to GLEAM as 'observational estimates' (L372) could be misleading.

We agree with AR1 that we need to be careful about how we refer to the GLEAM product. 'Observational estimates' in L391 has been changed to 'observations-based estimates' and other places in the text with similar language were also changed.

Additionally, a related point referring to the limitations of GLEAM was made by AR2 and has been addressed (see the document with our responses to AR2).

The results presented in Sect. 3.2 and Fig. 9 could benefit from difference maps and/or a quantification in terms of for example correlation, because the visual comparison of the maps is not entirely intuitive. Moreover, a short sentence on how to interpret the Theil-Sen slope could help the reader in understanding Fig. 9; what does a + and – slope mean in terms of water dynamics?

Thank you for these suggestions. We agree that quantification of the differences between plots in Fig. 9 would strengthen our results. We chose to calculate the correlation coefficient between plots, which requires regridding of the Noah-MP model results and the GLEAM data to a common spatial grid. Unfortunately, due to computation issues that arose in the last few days, we were unable to obtain the correlation coefficients before the response deadline. However, I am happy to provide them as soon as the issue is resolved and include them in the final version of the manuscript.

L243 – L248 in Methods section 2.4 have been updated to include more detail on the meaning of the Theil-Sen slope:

*We apply the Mann–Kendall test for monotonic trend (Mann, 1945; Kendall, 1948; Gilbert, 1987) to the mean time series of transpiration and ET between days 150 and 250 of the year (the height of the dry season) for each point in the domain. We then calculate the slope of the Mann–Kendall trend (known as the Theil-Sen estimator) for the mean time series for each point (Theil, 1950; Sen, 1968). The sign of the Theil-Sen estimator indicates the direction of the trend in the mean time series; a positive (negative) value is associated with generally increasing (decreasing) transpiration or ET. We compare spatial patterns of the Theil–Sen estimator between the SOIL and ROOT Noah-MP model runs and the gridded GLEAM evaporation estimates.*

Conclusion:

The conclusion provides more a summary of the findings than a strong conclusion. It is recommended that the authors rename the section, or reframe the section into a stronger ending. Furthermore, the numbers mentioned in L451 cannot be found in the previous sections, indicating that this does not fit in the conclusion section. It is also advised to stronger emphasize the relevance and the potential of this approach for the climate model community, and the accuracy of climate predictions.

The numbers mentioned in L451 (now L458-459) are found earlier in section 3.1. However, we agree with AR1 that the conclusion should include less details of the study

findings and more emphasis on the relevance of the study to the larger scientific community. We have reduced the detail included in the conclusion section considerably. Additionally, to improve the organization of the conclusion section, we have added subsection headings ('Summary of findings') and ('Significance and future work').

Elaboration on the significance of the study is done via addition of the following paragraph to the conclusion:

*Noah-MP with DynaRoot enabled can be used to investigate a number of different science questions with wide-ranging implications. Given the role of plant trait diversity in resilience of the Amazon as studied by Sakschewski et al. (2016) and the identification of deep-rooting as a drought resilience strategy by Chen et al. (2024), DynaRoot could be used to study changes in forest resilience under deforestation scenarios. Moreover, given that moisture varies slowly in subsurface soils (Amenu et al., 2005), DynaRoot makes it possible to characterize the role of deep soil moisture memory in influencing surface moisture via transpiration in a coupled land-atmosphere framework. Such research has been alluded to in Niu et al. (2020) and Zanin (2021), and could have implications for predictability of atmospheric moisture on longer timescales. Dominguez et al. (2024) discuss two multidecadal convection-permitting simulations that were completed for the entire South American continent. In their analysis of these runs, Zilli et al. (2024) identified land-atmosphere coupling in CPMs as an outstanding area of investigation. This motivates potential future work that focuses on the role of fine-scale land surface characteristics—such as water table depth and vegetation traits (including rooting depth)—in simulating convection. DynaRoot would be applicable in such work, particularly in global convection-permitting simulations that have become a priority in the climate modeling community (Satoh et al., 2019; Caldwell et al., 2021; Feng et al., 2023).*

**[Specific comments:**

- L100: reference brackets
    - o Reference brackets have been added.
- 1c,d: the colorbars of the soil texture and land cover types are lacking information. It is suggested that the authors either show only the classes that are present in the regions including labels, or all classes with all labels.
    - o Figure 1 has been updated to include only the classes that are present in the region for both soil texture and land cover.
- L242: 'uptake above 1m' is a bit confusing, could be solved by mentioning that we talk about uptake from soil layers shallower than 1m.

- o 'uptake above 1 m' has been changed to 'uptake shallower than 1 m'. This change has also been made to the caption and label of Fig. 2.
- L260: 'water table depth is 2m' would be more suitable
  - o In the GW case (unmodified Noah-MP with the MMF groundwater scheme activated), direct interactions between resolved soil layers and the water table can occur when the water table is shallower than the cumulative depth of all soil layers (in this case, 2 m). Thus, we will leave this part of L260 (now L276) as is.
- 3g: the units of the axes are missing and it is not explicit that this is for the ROOT experiment or for another experiment.
  - o Fig. 3g shows the root activity function (Eq. 9), which is unitless. This is now specified in the figure. The root activity function is part of the DynaRoot scheme, which is only active in the ROOT case. Thus, root activity output can only be from the ROOT case. The figure caption now specifies that the data in 3g comes from the ROOT case.
- L338: reference brackets
  - o Reference brackets have been added.
- L427: remove the 'is'
  - o Removed.

---

## Author Comment (AC2)

We thank Anonymous Referee #2 (AR2) for their helpful feedback on our manuscript. AR2's comments are copied below in black with our responses to each in red.

-Lines 49, 50 and 51: According to the Markewitz et al. (2010) study (cited), 10% of water uptake by roots occurs at depths between 550 cm and 1150 cm. So, correct this information! Moreover, instead of using only one modeling study for this information, use these two observational
studies: **https://doi.org/10.1002/hyp.6211** and **https://doi.org/10.1002/hyp.11143**

Thank you for this comment and for bringing the two observational studies to our attention.

L47-L51 has been updated:

*In a modeling study of an artificial throughfall exclusion experiment at Tapajós National Forest in northern Brazil (Nepstad, 2002; Nepstad et al., 2007; Davidson et al., 2011), Markewitz et al. (2010) noted that while the percentage of RWU occurring at depths between 5.5 and 11.5 m was relatively small (10%), model results suggest it was critical to survival.*

Directly afterwards, the following sentences have been added:

*Soil moisture observations collected by Bruno et al. (2006) in an Amazonian forest reflected withdrawal of soil moisture up to 10 m below the surface. Broedel et al. (2017) collected soil moisture observations from the central Amazon and found root uptake below 4.8 m during a year that was exceptionally dry.*

-Lines 85, 86 and 87: This sentence about the studies in Table 2 is wrong and should be corrected or deleted.

This sentence at L88-L90 has been updated:

*From Table 2, we see that representations of deep, dynamic RWU that do exist involve more complexity than needed for our purposes or do not include both deep and dynamic RWU.*

-Line 150: Mention here the depth at which most of the water uptake by the roots of Amazonian trees occurs (based on the two observational studies recommended previously).

This sentence at L152-L154 has been changed accordingly:

*This means hydrologically active soil layers in modified Noah-MP are deep enough to capture RWU consistent with uptake depths in the Amazon observed or inferred to be 4.8-18 m (Davidson et al., 2011; Bruno et al., 2006; Broedel et al., 2017).*

-Subsections 2.4 and 3.2: The GLEAM product consists of a set of algorithms to estimate the components of evapotranspiration, driven by satellite data. However, the maximum soil depth in this product is shallow (2.5 m). This should be mentioned as a limitation in these two subsections. Moreover, there are flux towers in the Brazilian state of Rondônia (with data freely available on the internet), and these should be considered.

The following has been added at L237-L238 in subsection 2.4:

*An important limitation of GLEAM is the soil module used in deriving the evaporation estimates, which includes shallow soil layers that only extend to 2.5 m (Martens et al., 2017). Despite this limitation, GLEAM is valuable in its temporal availability and partitioning of ET into components.*

Additionally, the following has been added at L388-389 in subsection 3.2:

*Keeping in mind the shallow soil module used to produce the GLEAM estimates, we note that the GLEAM values may be lower than if deeper soils were included.*

We are aware that flux tower data exists in Rondônia from the LBA project. However, data are only available for the early 2000s, coinciding with the years of model output we discarded to allow for spinup of the deep soil layers. This detail has been added to the manuscript at L239-L242:

*We note that flux tower data is available within our domain from the Large-Scale Biosphere-Atmosphere Experiment in Amazonia (LBA; Restrepo-Coupe et al., 2021). However, these data are only available for the early 2000s, coinciding with years that were discarded from the model output to account for spin up of deep soil layers.*

-Results Section: In the case of the Southern Amazon, it is more correct to refer to the austral summer (DJF) as the purely rainy season, and the austral winter (JJA) as the purely (and relatively) dry season. Or, to the period of the South American Monsoon as the rainy season (NDJFM), and the period completely outside this monsoon, as the relatively dry season (MJJAS).

The first paragraph of the Results section has been updated to reflect this comment:

*Figure 2 depicts simulation mean water table depth (WTD) and uptake shallower than 1 m in the ROOT experiment for all months, relatively dry months outside of the South American monsoon period (Jun-Sep), and relatively wet months during the monsoon period (Nov-Feb). Mean WTD is generally deeper in drier months (Fig. 2a), reflecting seasonal availability of moisture from precipitation. WTD is consistent with simulated values for the same region from other studies (Martinez et al., 2016a; Fan et al., 2017). Fractional uptake shallower than 1 m (Fig. 2b) varies between dry and wet periods, with a clear shift in uptake to depths below 1 m during drier months. This is consistent with a seasonal shift in RWU from shallower to deeper areas of the root zone as moisture from precipitation becomes scarce during drier months.*

The caption of Fig. 2 has also been updated in response to this comment.

-DISCUSSION SECTION:

--Lines 378 and 379: Mention that this refers to an offline model!

These lines (L407-L408) have been updated accordingly:

*The results of this work reflect a major advancement in the representation of the link between subsurface and atmospheric fluxes of moisture via RWU in the offline configuration of Noah-MP.*

--Line 380: Mention that it was evaluated in the southern Amazon.

The following has been added at L408-L409 :

*To demonstrate this, we focus on a region centered on the state of Rondônia in the southern Amazon.*

--Lines 393 and 394: Are you sure that the Amazon rainforest is water-limited?????

'Water-limited' has been removed from this part of the text.

--Overall: This subsection needs to be expanded and improved. One suggestion is that several studies that analyzed root depth and dynamics are mentioned in tables 1 and 2, and although some are simpler approaches than those in the present study, a comparative discussion of their results with the results of previous studies is important.

The following paragraph has been added to the Discussion section as the first paragraph. It includes text that was originally in the Conclusion, as well as a comparison of results from previous studies with our work:

*Overall, we find that the results of this work support the hypotheses detailed in the Introduction and are in accordance with previous studies that motivated these hypotheses. These include Fan et al. (2017), which highlighted the importance of groundwater as a moisture source for vegetation during dry periods, and Miguez-Macho and Fan (2021), which clarified that while moisture from groundwater is important in valleys, deep vadose zone storage of past precipitation is critically important in uplands during dry months. Additionally, the findings of this study are in line with others listed in Table 2, all of which found that inclusion of deep and/or dynamic RWU in Noah-MP improved model performance (Gayler et al., 2014; Wang et al., 2018; Liu et al., 2020; Niu et al., 2020; Li et al., 2021). In particular, Niu et al. (2020) and Li et al. (2021) noted improvements in Noah-MP's performance during drier periods after enhancements were made. Zanin (2021) is the only study in Table 2 that focused on the Amazon region and included the domain for this study. Similar to our work, they found changes in seasonality of soil moisture in shallow and deep layers resulting from addition of deep RWU. While simulation of sensible heat flux improved in Zanin (2021) when deep RWU was activated, latent heat flux was overestimated.*

-CONCLUSIONS SECTION:

--The first three paragraphs are a large summary of what was done in this article, and should be eliminated or simplified to a small paragraph.

Thank you for this comment. Based on this suggestion and a similar suggestion from AR1, we have removed many of the details from the first few paragraphs of the conclusion. Additionally, to improve the organization of this section, we have added two subsection headings: 'Summary of findings' and 'Significance and future work'.

--The fourth and fifth paragraphs should be placed in the discussion section.

These paragraphs include a summary of our findings, and as such, we leave them in the conclusion. However, as mentioned above, the organization of this section has been improved and the summary of findings has been reduced considerably. The intent of these changes is to improve the clarity of the discussion and conclusion sections. In consideration of this comment, the third to last paragraph was moved to the discussion section.

--The last three paragraphs are the only ones appropriate for the conclusions section, and should be explored further.

As mentioned above, the third to last paragraph was moved to the discussion section as we felt it made more sense to include it there. In response to this point as well as a similar comment from AR1, we added more detail to this section, including an additional paragraph that elaborates on avenues of research made possible by this study:

*Noah-MP with DynaRoot enabled can be used to investigate a number of different science questions with wide-ranging implications. Given the role of plant trait diversity in resilience of the Amazon as studied by Sakschewski et al. (2016) and the identification of deep-rooting as a drought resilience strategy by Chen et al. (2024), DynaRoot could be used to study changes in forest resilience under deforestation scenarios. Moreover, given that moisture varies slowly in subsurface soils (Amenu et al., 2005), DynaRoot makes it possible to characterize the role of deep soil moisture memory in influencing surface moisture via transpiration in a coupled land-atmosphere framework. Such research has been alluded to in Niu et al. (2020) and Zanin (2021), and could have implications for predictability of atmospheric moisture on longer timescales. Dominguez et al. (2024) discuss two multidecadal convection-permitting simulations that were completed for the entire South American continent. In their analysis of these runs, Zilli et al. (2024) identified land-atmosphere coupling in CPMs as an outstanding area of investigation. This motivates potential future work that focuses on the role of fine-scale land surface characteristics—such as water table depth and vegetation traits (including rooting depth)—in simulating convection. DynaRoot would be applicable in such work, particularly in global convection-permitting simulations that have become a priority in the climate modeling community (Satoh et al., 2019; Caldwell et al., 2021; Feng et al., 2023).*

---

## Referee Report (RR1)

I congratulate the authors for the improvements they have made. The article has significantly improved. I will only request two small adjustments. After these adjustments, I will recommend the article's publication. The author's responses are in red, and my comments are in blue.

The first paragraph of the Results section has been updated to reflect this comment:

*Figure 2 depicts simulation mean water table depth (WTD) and uptake shallower than 1 m in the ROOT experiment for all months, relatively dry months outside of the South American monsoon period (Jun-Sep), and relatively wet months during the monsoon period (Nov-Feb).*

*Mean WTD is generally deeper in drier months (Fig. 2a), reflecting seasonal availability of moisture from precipitation. WTD is consistent with simulated values for the same region from other studies (Martinez et al., 2016a; Fan et al., 2017). Fractional uptake shallower than 1 m (Fig. 2b) varies between dry and wet periods, with a clear shift in uptake to depths below 1 m during drier months. This is consistent with a seasonal shift in RWU from shallower to deeper areas of the root zone as moisture from precipitation becomes scarce during drier months.*

The caption of Fig. 2 has also been updated in response to this comment.

The correction was not done as I expected, but I accept. However, remove the mention of "South American Monsoon" or "monsoon" (the month of March is part of this monsoon, and you did not include it). Moreover, remove the word "relatively" before "wet months". Keep the word "relatively" only before "dry months".

The following paragraph has been added to the Discussion section as the first paragraph. It includes text that was originally in the Conclusion, as well as a comparison of results from previous studies with our work:

*Overall, we find that the results of this work support the hypotheses detailed in the Introduction and are in accordance with previous studies that motivated these hypotheses. These include Fan et al. (2017), which highlighted the importance of groundwater as a moisture source for vegetation during dry periods, and Miguez-Macho and Fan (2021), which clarified that while moisture from groundwater is important in valleys, deep vadose zone storage of past precipitation is critically important in uplands during dry months. Additionally, the findings of this study are in line with others listed in Table 2, all of which found that inclusion of deep and/or dynamic RWU in Noah-MP improved model performance (Gayler et al., 2014; Wang et al., 2018; Liu et al., 2020; Niu*

*et al., 2020; Li et al., 2021). In particular, Niu et al. (2020) and Li et al. (2021) noted improvements in Noah-MP's performance during drier periods after enhancements were made. Zanin (2021) is the only study in Table 2 that focused on the Amazon region and included the domain for this study. Similar to our work, they found changes in seasonality of soil moisture in shallow and deep layers resulting from addition of deep RWU. While simulation of sensible heat flux improved in Zanin (2021) when deep RWU was activated, latent heat flux was overestimated.*

Regarding the results of Zanin (2021) mentioned in the last sentence of this paragraph, it is important to mention that these results refer to the RJA flux tower. Moreover, it is interesting to mention that despite the overestimation of the simulated latent heat flux in this flux tower, the seasonality of simulated evapotranspiration is reduced with the deep soil water uptake by tree roots.

---

## Author Response (AR2)

We thank AR2 for their additional minor comments. Our responses to these comments are included in black below.

In addition to the changes suggested by AR2, some additional minor changes have been made, mostly to increase the clarity of the manuscript and correct minor errors. These additional changes are noted in the tracked changes document. We note that the equations in Section 2.2.2 have been updated to correct a minor error in the original manuscript. In these equations, $z_j$ is actually the midpoint of each soil layer, and is expressed relative to 1.5 m to ensure that soil properties do not vary in the first 4 shallow soil layers.
* * *
The first paragraph of the Results section has been updated to reflect this comment:

Figure 2 depicts simulation mean water table depth (WTD) and uptake shallower than 1 m in the ROOT experiment for all months, relatively dry months outside of the South American monsoon period (Jun-Sep), and relatively wet months during the monsoon period (Nov-Feb).

Mean WTD is generally deeper in drier months (Fig. 2a), reflecting seasonal availability of moisture from precipitation. WTD is consistent with simulated values for the same region from other studies (Martinez et al., 2016a; Fan et al., 2017). Frac onal uptake shallower than 1 m (Fig. 2b) varies between dry and wet periods, with a clear shift in uptake to depths below 1 m during drier months. This is consistent with a seasonal shift in RWU from shallower to deeper areas of the root zone as moisture from precipitation becomes scarce during drier months.

The caption of Fig. 2 has also been updated in response to this comment.

The correction was not done as I expected, but I accept. However, remove the mention of "South American Monsoon" or "monsoon" (the month of March is part of this monsoon, and you did not include it). Moreover, remove the word "relatively" before "wet months". Keep the word "relatively" only before "dry months".

Thank you for this feedback. This change has been made as requested.

The following paragraph has been added to the Discussion section as the first paragraph. It includes text that was originally in the Conclusion, as well as a comparison of results from previous studies with our work:

Overall, we find that the results of this work support the hypotheses detailed in the Introduction and are in accordance with previous studies that motivated these hypotheses.

These include Fan et al. (2017), which highlighted the importance of groundwater as a moisture source for vegetation during dry periods, and Miguez-Macho and Fan (2021), which clarified that while moisture from groundwater is important in valleys, deep vadose zone storage of past precipitation is critically important in uplands during dry months. Additionally, the findings of this study are in line with others listed in Table 2, all of which found that inclusion of deep and/or dynamic RWU in Noah-MP improved model performance (Gayler et al., 2014; Wang et al., 2018; Liu et al., 2020; Niu et al., 2020; Li et al., 2021). In particular, Niu et al. (2020) and Li et al. (2021) noted improvements in Noah-MP's performance during drier periods after enhancements were made. Zanin (2021) is the only study in Table 2 that focused on the Amazon region and included the domain for this study. Similar to our work, they found changes in seasonality of soil moisture in shallow and deep layers resulting from addition of deep RWU. While simulation of sensible heat flux improved in Zanin (2021) when deep RWU was activated, latent heat flux was overestimated.

Regarding the results of Zanin (2021) mentioned in the last sentence of this paragraph, it is important to mention that these results refer to the RJA flux tower. Moreover, it is interesting to mention that despite the overestimation of the simulated latent heat flux in this flux tower, the seasonality of simulated evapotranspiration is reduced with the deep soil water uptake by tree roots.

Thank you for this comment. This section of the manuscript has been updated:

Zanin (2021) is the only study in Table 2 that focused on the Amazon region and included the domain for this study. They compared their model results to flux tower data from the LBA project. Similar to our work, they found changes in seasonality of soil moisture in shallow and deep layers resulting from addition of deep RWU. While simulation of sensible heat flux improved in Zanin (2021) when deep RWU was activated, latent heat flux was overestimated. However, seasonality of evapotranspiration was reduced.